# Genetic Rescue of the Highly Inbred Norwegian Lundehund

**DOI:** 10.3390/genes13010163

**Published:** 2022-01-17

**Authors:** Claudia Melis, Cino Pertoldi, William Basil Ludington, Carol Beuchat, Gunnar Qvigstad, Astrid Vik Stronen

**Affiliations:** 1Department of Nature, Environment and Health, Queen Maud University College, Thrond Nergaards Veg 7, 7044 Trondheim, Norway; 2Department of Chemistry and Bioscience, Aalborg University, Fredrik Bajers Vej 7H, 9220 Aalborg Øst, Denmark; cp@bio.aau.dk (C.P.); astrid.stronen@gmail.com (A.V.S.); 3Aalborg Zoo, Mølleparkvej 63, 9000 Aalborg Øst, Denmark; 4Department of Embryology, Carnegie Institution for Science, Baltimore, MD 21218, USA; will.ludington@gmail.com; 5Department of Biology, Johns Hopkins University, Baltimore, MD 21218, USA; 6Institute of Canine Biology, Mission Viejo, CA 92691, USA; carol@instituteofcaninebiology.org; 7Department of Clinical and Molecular Medicine, Faculty of Medicine and Health Sciences, Norwegian University of Science and Technology (NTNU), 7491 Trondheim, Norway; gunnarq@ntnu.no; 8Department of Gastroenterology and Hepatology, St Olav’s Hospital, Trondheim University Hospital, 7006 Trondheim, Norway; 9Department of Biology, Biotechnical Faculty, University of Ljubljana, Večna pot 111, 1000 Ljubljana, Slovenia

**Keywords:** conservation breeding, domestic dogs, genetic diversity, native breeds, outcrossing, population recovery

## Abstract

Augmenting the genetic diversity of small, inbred populations by the introduction of new individuals is often termed “genetic rescue”. An example is the Norwegian Lundehund, a small spitz dog with inbreeding-related health problems that is being crossed with three Nordic breeds, including the Norwegian Buhund. Conservation breeding decisions for the (typically) small number of outcrossed individuals are vital for managing the rescue process, and we genotyped the Lundehund (*n* = 12), the Buhund (*n* = 12), their crosses (F1, *n* = 7) and first-generation backcrosses to the Lundehund (F2, *n* = 12) with >170,000 single nucleotide polymorphism loci to compare their levels of genetic diversity. We predicted that genome-wide diversity in F2 dogs would be higher than in the Lundehund but lower than in the F1 and the Buhund, and the heterozygosity values showed the expected patterns. We also found that runs of homozygosity, extended chromosomal regions of homozygous genotypes inherited from a common ancestor, were reduced in F2 individuals compared with Lundehund individuals. Our analyses demonstrate the benefits of outcrossing but indicate that some of the acquired genetic diversity is lost following immediate backcrossing. Additional breeding among F2 crosses could therefore merit from further consideration in genetic rescue management.

## 1. Introduction

### 1.1. Genetic Rescue in Domestic and Wild Populations

Increasing the genetic variability of small, isolated populations by the spontaneous or human-planned introduction of new individuals is often referred to as “genetic rescue” [1,2,3,4]. The positive effect of this management tool has been documented both in domestic and wild populations [1], although the long-term outcome can be difficult to evaluate [5], and some concerns exist that genetic rescue might lead to outbreeding depression [6]. Despite breeds of domestic dog (*Canis lupus familiaris*) being comparable to small and closed populations with a high frequency of breed-specific genetic disorders, genetic rescue by outcrossing with individuals of another breed is not a well-established practice among breeders. The main reasons for this are concerns about (1) contaminating lineage “purity”, (2) losing breed-specific phenotypic traits and (3) incorporating “new” genetic diseases from other dog breeds (unwanted introgression of deleterious alleles) [7].

Nevertheless, genetic disorders are likely to be less prevalent in a dog breed if a large, and therefore diverse, gene pool is maintained. Alleles with a negative effect on fitness are present in any population but only at low frequencies due to natural selection. However, when genetic variability is reduced, the frequency of such alleles will increase, whereby genetic disorders suddenly “appear” in the gene pool [7]. For this reason, it is challenging to manage genetic disorders by test screening in small populations. By excluding affected individuals from reproduction, we are also reducing the number of mating individuals (the effective population size) and, thus, the gene pool.

One important limitation of outcrossing programs for small populations, as revealed by computer simulations of different breeding schemes, is that outcrossing followed by backcrossing to the original population might only provide a short-term rescue effect, unless outcrossing is repeated continuously [7]. This is also supported by well-documented examples of genetic rescue in small and isolated wildlife populations, such as wolves (*C. lupus*) in Isle Royal and island foxes (*Urocyon littoralis*) [8].

Some examples of outcrossing in dog breeding are described on various web pages, although these efforts were initiated privately and are, as far as we know, not yet scientifically documented. For example, in 1973, the Pointer was crossbred to the Dalmatian by the geneticist and Dalmatian breeder Robert Schaible, because all Dalmatians worldwide tested positive for a metabolic illness called hyperuricosuria [9]. The first request to register a litter of Pointer × Dalmatian dogs to the American Kennel Club (AKC) was carried out in 1980, after five generations of backcrossing to Dalmatians to recover all breed-specific phenotypic traits. However, it took almost 30 years of controversy before more descendants of this backcrossing project could be registered to AKC [9]. Similarly, to avoid tail docking in Boxers, two Corgi × Boxer females were backcrossed to the Boxer in 1992 by the geneticist and Boxer breeder Dr. Bruce Cattanach [10], and a line of short-tailed (bobtail) Boxers that carry the dominant bobtail gene was developed in the United Kingdom.

In 2011, the Irish Kennel Club (IKC) established an outcrossing program between the Irish Red and White Setter and the Irish Setter and, in 2017, made a call to breeders worldwide to participate in the international outcrossing program to increase the genetic diversity of the Irish Red and White Setter [11]. However, it was only possible to access an archived copy of the announcement made by the IKC on the web [11], and, to our knowledge, no scientific publications or systematic documentation of the progress and results of this ongoing project have yet been made available.

### 1.2. Breed History and Typical Traits

The Norwegian Lundehund (hereafter Lundehund) is a small spitz dog traditionally used to hunt Atlantic puffins (*Fratercula arctica*, hereafter puffins) on steep cliffs along the northern coast of Norway. The breed is considered a cultural heritage and has several unique phenotypic traits, such as enhanced flexibility of the neck and shoulder joints, foldable ears and extra toes (polydactyly) on both front and back legs [12]. These peculiar traits might be the result of both natural and artificial selection, as they offer advantages when searching and retrieving puffins. The high flexibility of the joints would allow better mobility in narrow tunnels at the end of which puffins nest. Foldable ears would prevent dirt and parasites from entering the ears when working underground, and the combination of high flexibility and polydactyly would ensure better grip on the steep and loose terrain of the cliffs [13]. The small population suffered two bottlenecks in the 1940s and the 1960s, which resulted in the Lundehund being among the dog breeds with the highest reported inbreeding, as measured by microsatellite markers [12], single nucleotide polymorphisms (SNPs) [14,15] and genealogical data [16].

Due to a major effort to rescue the breed, today, there are about 1500 Lundehund individuals in the world, and about 900 of these live in Norway (https://natron.vm.ntnu.no/nlk accessed on 10 January 2021). Therefore, the breed is not at immediate risk of extinction.

However, the effective population size (N_e_) of the global Lundehund population was estimated to be very low based both on pedigree data [16] (N_e_ = 13) and on molecular data [17] (N_e_ = 28). Moreover, the Lundehund shows signs of inbreeding depression by reduced fertility due to small litter size, problems with mating behavior (probably due to inbreeding avoidance mechanisms) and low sperm quality [16]. Additionally, the Lundehund has a predisposition to develop intestinal lymphangiectasia, a protein-losing enteropathy that can cause symptoms such as intermittent diarrhea, vomiting, weight loss and ascites, often reported as the “Lundehund syndrome” (hereafter LS). In addition, chronic atrophic gastritis and gastric neoplasms are common in dogs with LS [14,15,17,18,19,20,21]. For this reason, breeders inform future owners about potential symptoms that could arise and require immediate veterinary treatment. Moreover, they advise new owners to feed their dogs with a diet low in fat content. Despite these preventive efforts, a study on mortality causes in the Lundehund conducted in 2010–2012 showed that 30% of deaths before 11 years of age occurred as a consequence of LS and another 10% of other gastrointestinal diseases [22]. Moreover, many dogs experience several acute episodes of LS throughout their lives, which require expensive treatment and negatively affect their quality of life. The pattern of inheritance of LS is not well understood [21] and might, at least in part, be explained by polygenic inheritance and a high frequency—or fixation—of the responsible gene(s).

### 1.3. The Outbreeding Project

These issues raised the question of whether it was ethical to continue breeding the Lundehund, and they motivated the Norwegian Kennel Club (NKK) to start an outbreeding project in 2014, which is still ongoing [16,23]. The aims of the project are to (1) increase the genetic variability of the breed, (2) improve fertility and (3) reduce the occurrence of LS in the population, while, at the same time, maintaining the unique traits of this breed [16,24].

Based on behavioral and morphological traits, shared history and genetic distance, three candidate breeds for outcrossing were selected by the NKK: the Norwegian Buhund (hereafter Buhund) (Figure 1b), the Norrbottenspets (from Sweden) and the Icelandic sheepdog [23].

The plan is to keep the crossings with these three breeds as three separate lineages with their own studbook and to monitor them for several generations before considering their inclusion in the Lundehund population and registering them in the main Lundehund studbook. All dogs used in the project were carefully selected based on health requirements established by the NKK, and each combination of Lundehund × Buhund also had to be approved by the NKK. Because the Buhund (12–18 kg) is larger than the Lundehund (6–9 kg), the first generation of crosses was made by mating a Buhund dam with a Lundehund sire. The Buhund dam should fulfil several requirements to be included in the project, such as to have grade A or B hip dysplasia (HD), grade 0 patella luxation (dislocated kneecap), be free of hereditary eye disorders, have a good temperament and generally have good health, confirmed by a health certificate. The same requirements, except HD status, applied to the Lundehund sire. In addition, both the Buhund and the Lundehund should have proven their fertility by having had litters before. These rather strict requirements, in addition to the difficulty in finding Buhund dam owners who were willing to let their dogs participate in the outcrossing project, limited the number of crossings that could be performed. The first two litters of the Lundehund × Buhund crosses were born in 2014, and their offspring was bred back to the Lundehund (Appendix A). All crosses are evaluated for good health, morphology and behavior at two years of age by a team of specialists, including a certified judge, before inclusion in the breeding project. The dogs are also checked for HD, patella luxation and hereditary eye diseases by a veterinarian officially recognized by NKK, as these conditions are present in the Buhund population. However, the Lundehund is not usually affected by these conditions, and NKK does not require any genetic test to breed them, which also allows a further reduction in the gene pool to be avoided. Individuals with serious behavioral issues (signs of fear or aggression) or health-related issues, such as monorchism or a severe degree of hip dysplasia, are excluded from the breeding project. Although individuals with light dysplasia are not given priority, these can nonetheless still be bred with healthy individuals. At this stage of the outbreeding project, the criteria for inclusion in breeding rely on health and behavior only and not on exterior appearance. The second-generation crosses produced by the project show all the specific traits, such as polydactyly, foldable ears and joint flexibility, whereas there is still a relatively large morphological variation in size, bone structure and ear shape. This variation would be rather normal in other dog breeds with a more diverse genetic pool, but with their reduced genetic variation, Lundehund individuals tend to show very low morphological variation. To date, none of the F1 and F2 individuals have shown signs of developing LS, although we should underline that the oldest F2 individuals are only five years old, and the success of the outbreeding project should be evaluated on a longer term. For the nine litters of F2 individuals that have been produced to date, the mean litter size was = 4.2 and the median was = 5, whereas for the Lundehund, both mean and median litter sizes were = 3 [25].

### 1.4. Aim of the Investigation

In this study, we investigated four groups of dogs, namely, Lundehund, Buhund and first- (F1) and second- (F2) generation Lundehund x Buhund crosses generated by the rescue project (Figure 1), to compare genetic diversity in the four groups. Although few outcrossed individuals are available for investigation so far, these dogs and, hence, their genetic profiles are vital for the rescue program. To allow comparison, the four groups were made of a similar sample size, where the F1 group had the fewest individuals due to the constraints described above. We predicted that the F2 dogs would show genome-wide diversity levels that were higher than those of Lundehund individuals but lower than those of the F1 and Buhund dogs. We assessed the four groups of dogs by combining genomic analyses of SNP profiles and data simulations.

## 2. Materials and Methods

### 2.1. Samples

Due to the non-invasive nature of DNA sampling (by buccal swabs), it was not necessary to apply for ethical approval of animal procedures to the Norwegian Animal Research Authority. In all cases, the collection of samples from individual dogs was approved by the dog owner. The following four dog groups were included in the study: Lundehund (*n* = 12), Buhund (*n* = 12), first-generation crosses Lundehund × Buhund (F1, *n* = 7) and first-generation backcrosses F1 × Lundehund (F2, *n* = 12). For an overview of the relatedness among individuals, see Appendix A. All dogs were DNA sampled with non-invasive buccal swabs, and the DNA was extracted with the Isohelix DDK-50 isolation kit. This sampling method is widely used and has been demonstrated to provide DNA of good quality, suitable for SNP studies [26]. The samples were genotyped with the Canine HD Bead Chip (Illumina) with 172,115 SNPs, and their quality was screened in GenomeStudio (Illumina) according to the program guidelines (http://www.illumina.com/Documents/products/technotes/technote_infinium_genotyping_data_analysis.pdf accessed on 10 January 2022). We screened individual profiles in PLINK v.1.90 [27] (https://www.cog-genomics.org/plink/1.9/ accessed on 10 January 2022) and retained dogs with individual and per-SNP call rates of >90%, resulting in 111,542 autosomal SNPs (genotyping rate > 0.99) for a total of 41 dogs. The profiles of two F2 dogs did not pass the screening process and were removed, resulting in *n* = 12 (6 females, 6 males) Lundehund, *n* = 12 (6 females, 6 males) Buhund, *n* = 7 (5 females, 2 males) F1 individuals and *n* = 10 (5 females, 5 males) F2 individuals. Next, we pruned the data for loci in linkage disequilibrium (LD), with a window size of 50 SNPs, a sliding window of 5 loci and a variance inflation factor threshold of 2 (PLINK command --indep 50 5 2), resulting in 8182 SNPs (henceforth the 8K dataset). Because the conservative LD pruning substantially reduced the number of SNP loci, we also performed a second, less stringent pruning that would retain more loci for assessment of runs of homozygosity (ROHs). The second dataset was obtained by filtering as described above but with pairwise genotype associations (r^2^) > 0.9 (--indep-pairwise 50 5 0.9 in PLINK), which retained 34,725 loci (henceforth 34K dataset).

### 2.2. Statistical Analyses

Genetic variability in each population was assessed by the calculation of observed heterozygosity (Ho); unbiased expected heterozygosity (uHE), an unbiased estimator of genetic diversity; the inbreeding coefficient (FIS); and the mean percentage of polymorphic loci (P) in GenAlEx 6.501 [28] (definitions and formulae are provided in Appendix A of the software manual). For HO, the skewness (a measure of the symmetry of a distribution), kurtosis (a measure of whether the data are heavy tailed or light tailed relative to a normal distribution), medians and 25% and 75% percentiles were calculated for every dog group. Because inbreeding will reduce heterozygosity and result in a skewed distribution with few loci showing high heterozygosity (a distribution with a long tail), this will produce high kurtosis (a peaked distribution). Because measures that assume a normal distribution in the data may perform less well when this condition is not met, it is relevant to monitor skewness and kurtosis [29,30], and these parameters can provide important information for the temporal analyses of genetic diversity.

The cumulative curves (representing the cumulative frequency distribution) for the Ho and uHE for every dog group were plotted for all the 8182 loci investigated, and the cumulative curves were compared among the groups.

We next investigated deviations from Hardy–Weinberg equilibrium (HWE) within each dog group with a statistical test in GENEPOP v4.3 [31].

Pairwise FST values were calculated for all combinations of dog groups to determine the degree of genetic differentiation using GenAlEx, and the Fisher’s exact probability test for genic differentiation was carried out using GENEPOP. We examined ROHs per group with the 34K dataset in PLINK (--homozyg–homozyg-group) with default parameter settings, and because of the small sample size, we included ROHs found in two or more individuals. We then plotted the results per autosomal chromosome for each of the four groups to visualize the differences among groups and among chromosomes.

## 3. Results

We first sought to validate the SNP data against the previous estimates of extremely low genetic diversity for the Lundehund and higher diversity for the Buhund. The mean Ho values for the four groups varied from 0.043 (Lundehund) to 0.272 (F1) (Table 1). The median Ho values were equal to zero for the Lundehund and the F2, indicating that more than 50% of the loci investigated were homozygotic, whereas the F1 and the Buhund showed a median value different from zero, indicating that more than 50% of the loci investigated were heterozygotes.

Only the Buhund showed a 25% percentile different from zero, illustrating that less than 25% of the loci were homozygotic in the Buhund. In contrast, only the Lundehund showed a 75% percentile equal to zero, indicating that more than 75% of the loci investigated were homozygotic in the Lundehund (Table 1).

The skewness values were all positive, ranging from 0.534 (Buhund) to 3.456 (Lundehund), whereas the kurtosis ranged from −4009.148 (Lundehund) to 680.202 (F1) (Table 1). The cumulative distribution curves of the heterozygosity for the 8182 variable loci showed clear differences among the different groups, with the Buhund showing the highest heterozygosity (as shown by the number of homozygous loci and by the curve, which exhibits the most gentle slope among the four groups). The Lundehund instead showed the lowest heterozygosity, and the F1 and F2 crosses showed intermediate values. It is noteworthy that, of the 8182 loci, the Buhund had less than 1000 homozygous loci, whereas the Lundehund had more than 7000 homozygous loci (Figure 2a,b).

Genetic variability parameters, including unbiased heterozygosity (uH_E_), inbreeding coefficient (F_IS_) and the mean percentage of polymorphic loci (P), are listed in Table 2.

Deviations from HWE were found to be highly significant (*p* < 0.001) for the F1, F2 and Lundehund and significant (*p* < 0.05) for the Buhund. All the deviations were due to heterozygote excess as can be seen by the negative F_IS_ values ranging from −0.420 (F1) to −0.051 (Buhund) (Table 2). The P ranged from 12.74% (Lundehund) to 90.96% (Buhund) (Table 2).

All the pairwise F_ST_ comparisons were highly statistically significant (*p* < 0.001), with values ranging from 0.055 (F1–F2 comparison) to 0.424 (Lundehund–Buhund comparison) (Table 3).

Runs of homozygosity (ROHs) indicate regions of the chromosome where a single genotype is contiguous. The ROH plot for the four dog groups showed, as expected, that the Lundehund had the highest number of ROHs, whereas no ROH was found in the first generation of crosses (F1) (Figure 3). Moreover, ROHs re-emerged in the next-generation F2, where they were more frequent and longer than those in the Buhund. The ROHs in the Lundehund included some long segments on chromosomes 9, 26 and 38 (Figure 3), and the number of ROHs per dog group was 247 for the Lundehund, 88 for the Buhund, none for the F1 and 116 for the F2.

## 4. Discussion

Our analysis suggests that the initial part of the Lundehund genetic rescue project has been successful, although further work remains to be carried out until outcrossed individuals can be officially included in the studbook of the Lundehund breed. In the F1 dogs, we observed a clear increase in unbiased heterozygosity (uHE) and the mean percentage of polymorphic loci (P) (Table 2). In the F2 generation, where the F1 dogs were backcrossed with Lundehund dogs, we observed the expected reduction in uHE and P compared to the F1s. These results illustrate the genetic rescue effect, as the uHE and P are higher in the F2 generation than in the parental Lundehund generation.

The cumulative heterozygosity plots quantify the increase in the number of heterozygous loci, which even reaches levels of uH_E_ and Ho that are higher than those of the Buhund generation (see Figure 2a,b), where the Lundehund genotype has enriched diversity over the Buhund in certain chromosomal regions. Additionally, the changes in the skewness and kurtosis of the genetic parameters reflect marked changes in the genomes across the generations. For example, the Lundehund’s strong positive Ho skewness (3.456) reflects an extremely depauperate genome with mostly homozygous loci. There are very few heterozygote loci in the tail of the distribution, i.e., with very low frequency, which illustrate the risk of an allele being lost, where the risk is inversely proportional to the allele’s frequency. Hence, several loci in the Lundehund are at risk of becoming fixed in a few generations.

Outcrossing with the Buhund considerably reduced the skewness in the F1 relative to that of the Lundehund (Table 1), although the value for the F1 was higher than that of the Buhund. It is noteworthy that if, instead of the median, we had used only the mean Ho for monitoring the changes in Ho across generations, we would have observed a minimal change in the F1 compared to the Buhund (Table 1). This discrepancy is a consequence of the skewed distributions of the Ho. A higher skewness reflects an increase in the number of heterozygote loci, thus reducing the number of loci at risk of becoming fixed, which was the main reason for initiating the genetic rescue project. The highly skewed distributions of the genetic parameters also suggest that their medians are highly informative and complementary to the mean values. The use of mean values could otherwise be misleading, as these only provide estimates for the central value of a distribution if the distribution is symmetrical around the mean. For skewed distributions, the median is therefore a better descriptor of a distribution’s central value [29,30].

In the F2 generation, the skewness increased relative to that of the F1 generation (1.402 versus 0.956), because many loci that were heterozygotes in the F1 became homozygotes when backcrossing with Lundehund dogs to create the F2. If we had used the mean Ho to compare the F2 and F1, we would have observed a reduction of 0.272 − 0.153 = 0.119, which is equivalent to a reduction of 43.75%, but the medians show a more dramatic scenario with more than 50% of homozygous loci. However, there are fewer homozygous loci in the F2 compared to the Lundehund, even if the 75% upper quantile in the F2 was reduced compared to the F1 (Table 1).The kurtosis values were not very informative in this investigation, as their values were strongly influenced by the strong asymmetry of the distributions of the Ho values.

Although genetic rescue efforts have provided successful results for various species and populations [1], additional gene flow may be needed to ensure persistence over the long term [5,32]. Despite uH_E,_ P and F_IS_ being increased in the F2, we can clearly see that backcrossing the F1 with the Lundehund reduced the genetic distance between the F2 and the parental Lundehund (F_ST_ = 0.134) compared to the distance between the F1 and the parental Lundehund (F_ST_ = 0.319). A future backcross of the F2 with the Lundehund could further reduce the genetic distance between the next generation (F3) and the parental population and result in an additional loss of genetic variability.

The ROH results reflect the genetic rescue effect by illustrating the differences between the Lundehund genome and that of the other groups. However, the immediate backcrossing with the F1 and the Lundehund to form the F2 resulted in the rapid re-emergence of several ROH segments. This also raises the question of whether additional crossing between F2 individuals or the possible crossing of F2 individuals from different types of crossings (Lundehund × Buhund, Lundehund × Icelandic sheepdog and Lundehund × Norrbottenspets) could present alternative scenarios for preserving genetic diversity and reducing homozygosity in future generations.

## 5. Conclusions

This study clearly documents the beneficial genetic effect of outcrossing a highly inbred dog population. It also documents that backcrossing the F1 generation to the parental population results in a loss of some of the desired heterozygosity achieved in the initial outcross. To preserve the characteristics of the Lundehund, the F2 dogs were made by crossing the F1 with the Lundehund and not with other F2 dogs from different lineages. While subsequent backcross of the F1 with the Buhund would have produced an F2 generation with higher levels of heterozygosity than those achieved by the F1 × Lundehund cross, our study quantifies the effect that the immediate backcross strategy had on the genetic diversity of the F2 dogs. Thus, stakeholders in the program and in future genetic rescue projects can use this data in combination with health data for the F1 and F2 animals to evaluate the effectiveness of the genetic rescue program so far. Our results indicate that additional crossbreeding would extend and augment the genetic rescue process. These include crossing among F2 dogs from the three different breeds involved in the outcrossing project (Buhund, Norrbottenspets and Icelandic sheepdog) to further increase and maintain genetic diversity. A careful evaluation of the resulting phenotypes, particularly with respect to LS [15,16,17,18,19], could help identify the genes involved in the disease and, thus, allow selection of the desired variants for the breeding program.

## Figures and Tables

**Figure 1 genes-13-00163-f001:**
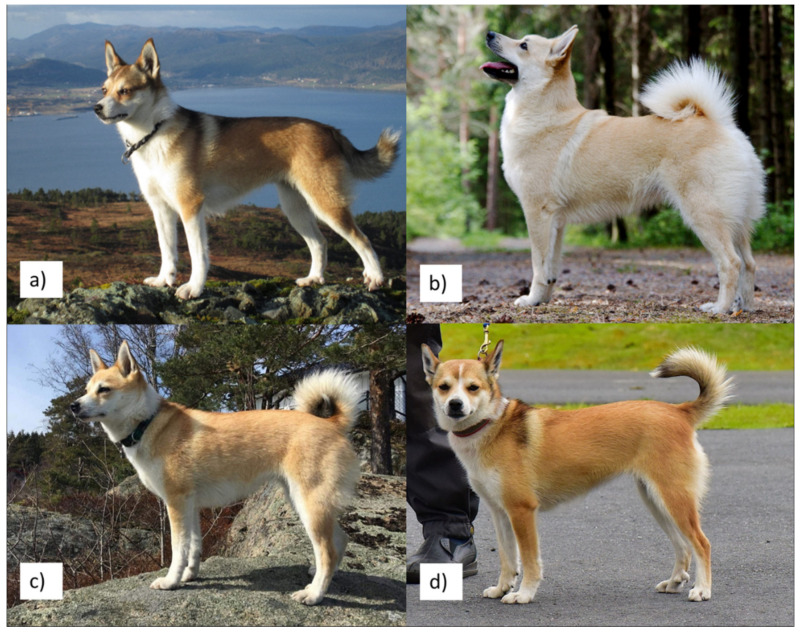
Dog breeds used in the outbreeding project and their descendants. (**a**): Norwegian Lundehund; (**b**): Norwegian Buhund; (**c**): F1 first-generation crossing of Lundehund × Buhund; (**d**): F2 second-generation crossing of F1 × Lundehund. All individuals are females. (**a**): Photo by Dagrunn Mæhlen, (**b**): photo by Ina Margrethe Gabrielsen Egren, (**c**): photo by Cathrine Brekke, (**d**): photo by Arild Espelien.

**Figure 2 genes-13-00163-f002:**
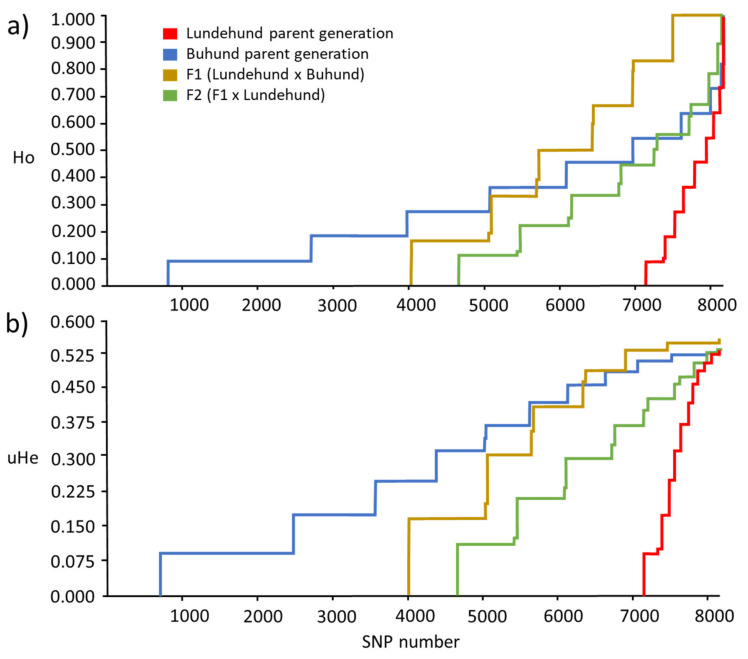
Cumulative curves of the observed heterozygosity (Ho) (**a**) and unbiased expected heterozygosity (uH_E_) (**b**) estimated from 8184 loci of the 4 dog groups: Lundehund, Buhund, first-generation crosses F1 (Lundehund × Buhund) and first-generation backcrosses F2 (F1 × Lundehund). The lines show a decline in the number of loci with a certain range of Ho and uHe (the lengths of the horizontal lines) per generation. In the parental generation, the Buhund has the longest horizontal lines followed by the F1 generation, the F2 generation and, finally, the Lundehund.

**Figure 3 genes-13-00163-f003:**
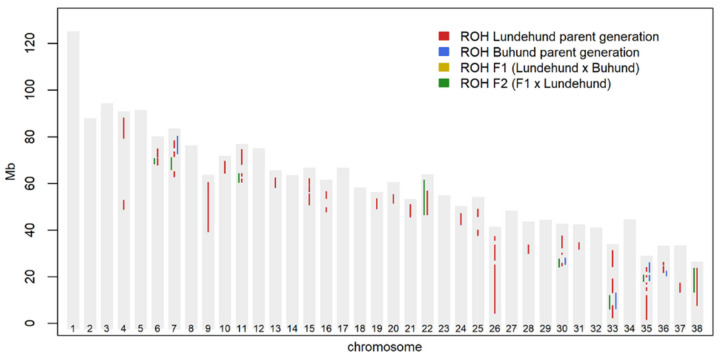
Plots of runs of homozygosity (ROHs) per breed and per chromosome, showing ROHs shared by two or more individuals and based on 34K SNP loci. ROHs were relatively frequent in the Lundehund, and some were also detected in the Buhund, whereas none were observed in the F1 generation. Although we did not sample all the Lundehund individuals involved in the rescue project, the results show that the process has provided, at least temporarily, an increase in genetic diversity.

**Table 1 genes-13-00163-t001:** Summary of the observed heterozygosity (Ho) (estimated from 8184 linkage-disequilibrium-pruned loci) from each of the 4 dog groups: Lundehund (LUN), Buhund (BUH), first-generation crosses LUN × BUH (F1) and first-generation backcrosses F1 × Lundehund (F2). The number of individuals sampled in each group (n), the mean Ho, the standard error of the mean (S.E.), the median Ho, the 25% and 75% percentiles, the skewness and the kurtosis of the Ho distributions are shown.

	LUN Ho	BUH Ho	F1 Ho	F2 Ho
n	12	12	7	10
Mean	0.043	0.269	0.272	0.153
S.E.	0.001	0.002	0.004	0.002
Median	0	0.273	0.167	0
25%	0	0.091	0	0
75%	0	0.455	0.5	0.25
Skewness	3.456	0.534	0.956	1.402
Kurtosis	−4009.148	−659.838	−680.202	−1108.418

**Table 2 genes-13-00163-t002:** Indices of genetic diversity per group: Lundehund (LUN), Buhund (BUH), first-generation crosses LUN × BUH (F1) and first-generation backcrosses F1 × LUN (F2). The table presents the unbiased expected heterozygosity (uH_E_) and inbreeding coefficient (F_IS_) and their respective standard errors, the Hardy–Weinberg test (HWE) and the mean percentage of polymorphic loci (P).

Group	uHE ± SE	F_IS_ ± SE	HWE Test	P
LUN	0.041 ± 0.001	−0.083 ± 0.003	***	12.74%
BUH	0.267 ± 0.002	−0.051 ± 0.003	*	90.96%
F1	0.195 ± 0.002	−0.420 ± 0.004	***	50.89%
F2	0.127 ± 0.002	−0.216 ± 0.002	***	42.93%

*** *p* < 0.001, * *p* < 0.05.

**Table 3 genes-13-00163-t003:** Pairwise F_ST_ values (upper diagonal) of the four dog groups: Lundehund (LUN), Buhund (BUH), first-generation crosses LUN × BUH (F1) and first-generation backcrosses F1 × LUN (F2). All F_ST_ comparisons were highly statistically significant (*p* < 0.001).

	BUH	F1	F2
LUN	0.424	0.319	0.134
BUH		0.1241	0.252
F1			0.319

## Data Availability

The data presented in this study are available on request from the corresponding author. The data are not publicly available because they are part of an ongoing study.

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
