# Peer review of "Genetic Rescue of the Highly Inbred Norwegian Lundehund"

_genes, 2022, doi:10.3390/genes13010163_

Round 1

Reviewer 1 Report

The manuscript entitled “Genetic rescue of the highly inbred Norwegian Lundehund” describes changes of genetic diversity during the outcrossing of the two dog breeds followed by backrosses to one of them, using SNP data. Overall, the topic is very important and deserves attention. It's really great to see that the breeding club is using modern techniques to evaluate their breeding management.

In the introduction, the authors are pointing to threats that always have to be considered during such experiments as genetic rescue, however, the whole manuscript deals only with descriptive parameters that do not give a clue about the viability of the population and the overall genetic diversity is considered as the only merit that matters - which is of course not necessarily the true. The whole manuscript does not bring a wider picture of the experiment and described results are predictable and do not bring new knowledge to the topic. The manuscript would benefit from including the other collected data, such as phenotypic or behavioural traits. 

Used analytical tools are standard and of a good quality, however, the authors are using very low sample sizes which should be emphasised in the text as one of the limits of the study (at least in the purebred individuals). I am also missing comparison with other breeds that are local with even smaller population size and were rescued through outcrossing and later restoration. There are many examples. 

L54-62 1500 individuals (are those breeding individuals?) are probably not representing small population sizes, there are breeds with smaller population sizes where genetic screening is a powerful and effective tool, helping to prevent display of hereditary diseases.

I recommend using standardized positions of the dogs on the pictures at Figure 1. If possible, use the same posture as visible on Fig. 1d

Reviewer 2 Report

Authors genotyped Lundehund, Buhund, their crosses (F1) and first-generation backcrosses to Lundehund with > 170 000 single nucleotide polymorphism loci to compare their levels of genetic diversity. This research is important for the conservation of the Lundehund.

  1. Line 76. “The small population suffered two bottlenecks in the 40s and the 60s”. “the 40s and the 60s” should be changed to “the 1940s and the 1960s”.
  2. Line 148. In this experiment, the DNA samples were collected by buccal swabs. Whether the method is reasonable to obtain the DNA for dogs, without contamination by other bacteria? The authors should give the reference to prove this method is reasonable.
  3. Line 151-153. The gender of the dogs should be added.
  4. Line 170. “(r2) > 0.9”. The “2” should be superscript.
  5. Line 190. Because of the small number of the samples, whether is necessary to test the Hardy-Weinberg Equilibrium (HWE)?
  6. Line 220-221. The authors stated that “with Buhund showing the highest heterozygosity”. According to Figure 2a, the description was not consistent with the results. In Figure 2a, F1 showed the highest heterozygosity for the 8182 variable loci.
